# Cardio Protective Effects of Lipid Emulsion against Ropivacaine-Induced Local Anesthetic Systemic Toxicity—An Experimental Study

**DOI:** 10.3390/jcm11102784

**Published:** 2022-05-15

**Authors:** Alexandra Elena Lazar, Simona Gurzu, Attila Kovecsi, Marcel Perian, Bogdan Cordos, Mircea Constantin Gherghinescu, Liviu Sorin Enache

**Affiliations:** 1Department of Anesthesiology, Emergency Clinical County Hospital, University of Medicine, Science and Technology “George Emil Palade”, 540136 Tirgu Mures, Romania; alexandra.lazar@umfst.ro; 2Department of Morphopathology, Emergency Clinical County Hospital, University of Medicine, Science and Technology “George Emil Palade”, 540136 Tirgu Mures, Romania; kovecsiattila@gmail.com; 3Department of Physiology, University of Medicine, Science and Technology “George Emil Palade”, 540136 Tirgu Mures, Romania; marcelperian@gmail.com; 4Veterinary Experimental Base, University of Medicine, Science and Technology “George Emil Palade”, 540136 Tirgu Mures, Romania; exoticvet@yahoo.com; 5Department of Surgery, Emergency Clinical County Hospital, University of Medicine, Science and Technology “George Emil Palade”, 540136 Tirgu Mures, Romania; dr.gherghinescu@yahoo.com; 6Emergency Clinical County Hospital Tirgu Mures, University “Dimitrie Cantemir”, 540136 Tirgu Mures, Romania; liviusenache@yahoo.com

**Keywords:** local anesthetic systemic toxicity, ropivacaine, lipid emulsion, protective effect, morphopathology

## Abstract

Inadvertent intravascular injection of local anesthetics (LA) during regional anesthesia causes Local Anesthetic Systemic Toxicity (LAST). Theories of lipid rescue in the case of LAST proved that the administration of lipids in LAST has beneficial effects. One possible mechanism of action is based on the lipophilic properties of LA which allow plasma-free LA to be bound by the molecules of Lipid Emulsion (LE). The association LA–LE is shuttled towards organs such as liver and the kidneys, and the half-life of LA is shortened. The main objective of this experimental study was to assess the possible cardio-prophylactic effect of LE administration before the induction of LAST by intravenous administration of Ropivacaine. This was an experimental, interventional, prospective, and non-randomized study. The subjects were divided into groups and received, under general anesthesia, LE 20% first 0.3–0.4 mL, followed by 0.1 mL Ropivacaine 2 mg/mL, or Ropivacaine alone. At the end of the experiment, the subjects were sacrificed, and tissue samples of kidney, heart and liver were harvested for histopathological examination. LE, when administered as prophylaxis in Ropivacaine-induced LAST, had protective cardiac effects in rats. The LE known side effects were not produced if the substance was administered in the low doses used for LAST prophylaxis.

## 1. Introduction

Local anesthetics (LA) are a class of pharmacological agents widely used in current medical practice, ranging from small surgical interventions to more complex procedures, such as wound sutures, or reconstructive hand surgery. The inadvertent intravascular injection of LA during regional anesthesia causes Local Anesthetic Systemic Toxicity (LAST). This syndrome includes seizures, malignant arrhythmias to cardiac arrest, hardly responsive to resuscitation [1].

These modifications might affect the patient’s sensory and vision and determine muscular stimulation with convulsions. Through the negative influence they have upon the resting membrane potential, LA affects the normal conduction within the heart cells at the cardiac level. The corresponding EKG modifications are prolonged PR, ORS, and ST intervals, re-entrant tachyarrhythmias, and bradyarrhythmia leading to a prolonged QR interval [2]. The mechanism by which the LA produces this syndrome is the Na channel blockage maintaining the channels in the inactive state. In this state, the Na channels cannot respond to stimuli, and the transmission of the action potential is impeded, thus the inhibitory neuron depolarization is also obstructed [3] (Figure 1).

Although the occurrence of LAST is not very frequent, with a number of 459 cases being reported in a 5-year period, by the National Poison Data System (NPDS)-USA, the catastrophic manifestation of this event - seizures, malignant arrhythmias, cardiac arrest resistance to resuscitative measures - maintains the severity of this condition [4].

In the past two decades, theories of lipid rescue in the case of LAST emerged and proved that the administration of lipids in LAST has beneficial effects [5,6,7]. Moreover, anesthesiologists included lipid emulsion (LE) in their recommendation for LAST treatment [8].

Although the exact mechanism by which lipids are useful in the case of LAST is not fully understood, some cases of LAST were successfully treated by LE administration during the toxic event [6,9]. (LA) are water-soluble salts of lipid-soluble alkaloids. The structure of local anesthetics comprises a lipophilic aromatic group and a hydrophilic amino group, bound by an intermediary link [10]. One of the possible mechanisms of action is based on the lipophilic properties of LA which allow plasma-free LA to be bound by the molecules of LE. The association LA–LE is shuttled towards the organs such as the liver and kidneys, and the half-life of LA is shortened [11]. LE is known to have beneficial cardiac effects in LAST by increasing cardiac contractility and, consequently, cardiac output [12].

Among LE side effects, the ones worth mentioning are acute kidney injury, metabolic acidosis, acute lung micro embolism, or cardiac arrest [13,14]. The incidence of the LE complications is listed in the manufacturer’s marketing release as less frequent (less than 1%) [15].

The main objective of this experimental study was to assess the possible cardio-prophylactic effect of LE administration before the induction of LAST by intravenous administration of Ropivacaine.

Secondly, we aimed to observe whether lipids, prophylactically administered, induce histological changes upon heart, liver, and kidney.

## 2. Materials and Method

The study was approved by the Ethics Commission of the University of Medicine, Pharmacy, Science and Technology “George Emil Palade” of Targu-Mures, Tirgu Mures, Romania (Approval No. 82/2019).

This was an experimental, interventional, prospective, and non-randomized study.

The study comprised Wistar male rats aged 10–12 weeks and weighing between 200 and 300 g.

The subjects were divided into 4 groups:

Group A—(*n* = 10)—LA Ropivacaine 2 mg/mL 0.1 mL was administered every 2 min and cardiac activity was monitored after 4 administrations. Tissue samples were harvested from the kidney, heart, and liver.

Group B—(*n* = 9)—LE 20% 0.3–0.4 mL was administered 2 min prior to 0.1 mL Ropivacaine 2 mg/mL. Cardiac activity was monitored and, after 4 administrations of Ropivacaine, the subject was sacrificed, and tissue samples were harvested from the kidney, heart, and liver.

Group C—(*n* = 8) which only received Lipid Emulsion 20% under general anesthesia. Cardiac activity was monitored and tissue samples from kidney, heart, and liver were harvested.

Group D—control group (*n* = 8) which received none of the studied substances. Subjects were anesthetized, the cardiac activity was monitored and tissue samples from kidney, heart, and liver were harvested.

The subjects were acclimated to standard laboratory conditions for 14 days, respecting the circadian rhythm, and free water and food intake access.

Four hours before the experiment, the subjects were fasted for water and food. The experiments were performed entirely under general anesthesia, using Isoflurane (Anesteran, ROMPHARM COMPANY, Bucharest, Romania). The studied substances—Ropivacaine 2 mg/mL concentration 0.1 mL repeated at 2 min intervals (Fresenius Kabi, Bad Homburg, Germany) and Lipid Emulsion 20% (Intralipid^®^ Fresenius Kabi, Bad Homburg, Germany) 0.3–0.4 mL 2 min prior to Ropivacaine—were administered via inferior vena cava puncture. The dosages were adjusted to weight. The invasive maneuvers and the exact dosage calculation were performed as previously described by the authors Lazar, A. et al. Acta Medica 2021; 67(2) 90–94 [16]. According to our primary objective, the heart rate and rhythm of subjects in groups A and B were monitored using LabView software.

The cardioprotective effect of LE 20% was assessed by rhythm analysis of the cardiac tracings of the subjects. A protective effect was considered when the normal cardiac rhythm was maintained longer or the EKG components (length of the QRS complexes or the heart rate) were maintained after Ropivacaine was administered.

At the end of the experiment, the subjects were sacrificed by exsanguination and tissue samples from the heart, liver, and kidney were harvested for histopathological assessment. Groups C and D were included for the secondary objective, in the assessment of possible histological changes induced by LE in various organs. Electrophysiological investigations were not performed in these groups.

## 3. Histopathological Assessment

The tissue samples from the heart, liver, and kidney were preserved in neutral buffered formaline solution 10%, embedded in paraffin, and stained with Hematoxylin Eosin. Two experienced pathologists (SG, AK) examined the possible histological changes of the examined tissues, in a blind manner.

## 4. Statistical Analysis

Data were centralized in Microsoft Excel spreadsheets and analyzed in the R statistical environment (version 4.1.1, cran.r-project.org, accessed on 1 September 2021). Normal distribution of quantitative data was verified using the Shapiro–Wilk test, whereas statistical hypothesis testing regarding the differences in biological parameters between the experimental groups was performed using Student’s two-sample *t*-test for paired/unpaired data, as required. A statistical significance threshold of 0.05 was considered.

## 5. Results

### 5.1. Clinical Data

To explore the possible protective role of lipid emulsion against the cardiac toxicity due to systemic administration of LA, rats were randomly assigned into four groups: in the first group, only intravenous LA Ropivacaine was administrated (group A; *n* = 10); in the second group, intravenous LE 20% was used before Ropivacaine (group B; *n* = 9). The third group (group C; *n* = 8) received only LE 20%, and the fourth (group D; *n* = 8) was the control group. The baseline values of cardiac activity in groups A and B were similar for both heart rate (*p* = 0.073) and QRS duration (*p* = 0.715) (Table 1).

In rats from group B, intravenous LE did not change either the heart rate (*p* = 0.085, paired *t*-test) or the QRS interval duration (*p* = 0.129, paired *t*-test).

Intravenous administration of Ropivacaine significantly reduced heart rate (*p* < 0.001) and prolonged QRS duration (*p* < 0.001) in both experimental groups (Figure 2). However, the heart rate decrease was much more pronounced in group A, with a relative change of −57% (SD = 25.4%), compared with group B, with a relative change of only –28.5% (SD = 9.94%) (*p* = 0.006, *t*-test). The relative change of QRS duration was similar in the two groups (195%, SD = 148% in group A, vs. 134%, SD = 65.8% in group L + A, *p* = 0.262, *t*-test).

### 5.2. Microscopy Findings

In the samples from heart, liver and kidney, no histological changes were seen, compared with the control group, for cases from group A or B (Figure 3).

## 6. Discussion

Regional anesthesia/analgesia techniques are increasingly employed for surgery or for postoperative analgesia to reduce opioid administration in the so-known Opioid Free Anesthesia/Analgesia (OFA) current [17].

Regardless of the safety measures taken to avoid LAST—ultrasound usage, echogenic needles, aspiration before injecting the LA—accidents can happen, and the LA can reach the systemic circulation.

There are several theories about how LE helps in LAST, none of them concluding an exact mechanism of action. The most popular is the “lipid sink” theory, which is based on the liposolubility of local anesthetics [18]. Although in many cases the LE proved to be of real help in LAST resuscitation, the idea of this study is based on the fact that to prevent is better than to treat [6,10].

To induce LAST, we used Ropivacaine, a relatively recent LA first produced as a pure enantiomer. Ropivacaine is a long-acting amide LA extensively metabolized by the liver [19].

To our knowledge, this is one of the few experimental studies which evaluated the protective effects of LE in Ropivacaine-induced LAST.

The results support the hypothesis that LE has beneficial effects against the cardiac manifestations of LAST.

The maintenance of heart rate and QRS duration close to baseline range for a longer period in rats prophylactically treated with LE prior to LA administration suggests the cardioprotective effects of LE. Our results are concordant with those from other studies [20,21].

The histopathology samples of the main organs showed that LE administered in the low dose used as a prophylactic measure in this study does not induce any organ abnormalities, at least in the immediate term. Along with the already-known safety of LE administration, our observations support the introduction of LE as prophylaxis before regional anesthesia, to reduce even more the risk of cardiac arrest caused by a possible inadvertent intravascular administration of LA.

This experimental study has several limitations. One of them is the unknown plasma level of LE or the LA, although the administration was according to international guidelines for lipid rescue in LAST. However, the time interval between LE and LA administration in our setup (2 min) was much shorter than the half-life of long-chain triglycerides, the main constituent of Intralipid administered in rats, which was reported to be more than 30 min [22]. We can thus safely assume that most of the LE was still present in the circulation at the time Ropivacaine was injected.

The next phase of this study contains the prospect of LE and LA plasma level measurement. In the phase presented herein, we aimed to observe the influence upon clinical signs.

Another limitation that can be considered is the use of general anesthesia in our subjects, given the available evidence in the literature that neurological manifestations in LAST appear earlier, and at lower doses of LA, than the alteration of cardiac function. In our setup, we only explored the case wherein the LA concentration was high enough to trigger anomalies in heart activity, while the potential neurological impairment could not be assessed.

These limitations will be addressed in future experimental models.

In present clinical practice, LE is only used in LAST as a treatment after the onset of symptoms. The novelty of our study is the introduction of LE as a prophylactic agent in Ropivacaine-induced LAST, instead of a treatment for an already-diagnosed toxicity. Although not sufficient for supporting the routine prophylactic use of LE in all procedures requiring LA, our results are encouraging, showing a cardioprotective effect and negligible toxicity of LE in our experimental model. Further safety and efficacy studies are warranted to explore the benefit of this prophylactic approach.

## 7. Conclusions

Lipid Emulsion, when administered as prophylaxis in Ropivacaine-induced LAST, has protective cardiac effects in rats, illustrated by sustaining the heart rate and QRS duration for a longer period.

The Lipid Emulsion known side effects are not produced if the substance is administered in the low doses required for LAST prophylaxis.

## Figures and Tables

**Figure 1 jcm-11-02784-f001:**
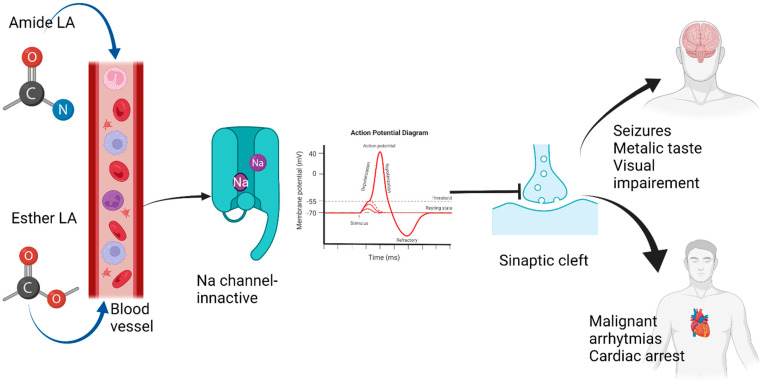
Local anesthetic systemic toxicity mechanism and effects upon the brain and heart.

**Figure 2 jcm-11-02784-f002:**
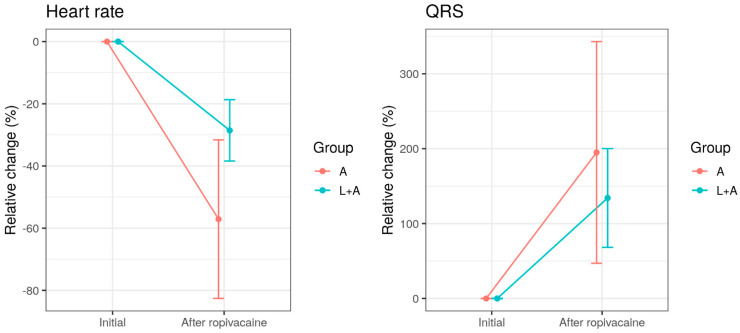
Administration of lipid emulsion prior to Ropivacaine (L + A) does not prove to induce relative change of heart rate and QRS complex duration in rats exposed to intravenous Ropivacaine only (A).

**Figure 3 jcm-11-02784-f003:**
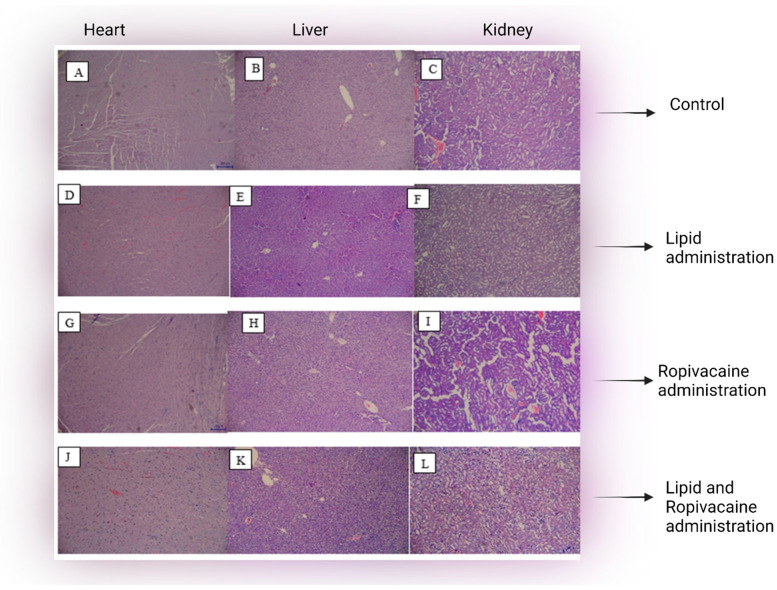
Microscopic assessment of tissue samples from heart, liver and kidney: panels (**A**–**C**)—control; panels (**D**–**F**)—after administration of lipid emulsion only; panels (**G**–**I**)—after local anesthetic (Ropivacaine) administration; panels (**J**–**L**)—after administration of lipid emulsion prior to local anesthetic. No histological changes were seen among groups, in any of the examined organs, after administration of lipid, local anesthetic, or both.

**Table 1 jcm-11-02784-t001:** Cardiac response to intravenous ropivacaine preceded or not by administration of lipid emulsion.

	Group A *n* = 10	Group B *n* = 9
*Baseline cardiac parameters*		
Heart rate (min^−1^)	380 (66.0)	321 (68.3)
QRS duration (ms)	20.6 (5.44)	19.7 (5.48)
*Cardiac parameters after intravenous Lipid emulsion*		
Heart rate (min^−1^)	-	296 (78.4)
QRS duration (ms)	-	22.9 (4.26)
*Cardiac parameters after intravenous Ropivacaine*		
Heart rate (min^−1^)	156 (93.2)	226 (37.5)
QRS duration (ms)	55.6 (21.1)	45.0 (14.4)

- data presented as mean (standard deviation).

## Data Availability

Authors of this article are in possession of the data sets and these can be released upon request.

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
