# Peer review of "Cardio Protective Effects of Lipid Emulsion against Ropivacaine-Induced Local Anesthetic Systemic Toxicity—An Experimental Study"

_jcm, 2022, doi:10.3390/jcm11102784_

Round 1

Reviewer 1 Report

Major Comments:

  1. The goal of this study was to investigate cardiac protective effects of lipid emulsion prior to the administration of local anesthesia.  The role of lipid emulsion in treating LAST is well accepted and understood in the anesthesia community. The authors state that this study is novel in that it is investigating the prophylactic effect of LE, however it is not clear how early LE is administered prior to LA in Group B. If the goal is to understand the prophylactic effects of LE It would be beneficial for the authors to investigate the half-life and plasma concentration of LE after administration.

Minor Comments:

  1. The difference in HR between the two groups was trending significance. The authors should comment on this and perhaps improve the study design by increasing the number of rats studied to diminish this difference between groups
  2. Similarly, the decrease in HR due to LE in Group B trended statistical significance. An increase in number of rats studied would help clarify this result.

Author Response

Responses to Reviewer #1.

Thank you very much for your interest in our manuscript and for your helpful observations. Please find our point-by-point responses to your comments and suggestions.

“Major Comments:

  1. The goal of this study was to investigate cardiac protective effects of lipid emulsion prior to the administration of local anesthesia. The role of lipid emulsion in treating LAST is well accepted and understood in the anesthesia community. The authors state that this study is novel in that it is investigating the prophylactic effect of LE, however it is not clear how early LE is administered prior to LA in Group B. If the goal is to understand the prophylactic effects of LE It would be beneficial for the authors to investigate the half-life and plasma concentration of LE after administration.”

We clarified the timing of LA and LE administration, in the Methods Section of the manuscript. More precisely, in group B, LE 20% was administered 2 minutes prior to 0.1ml Ropivacaine 2mg/ml. We acknowledge the limitations of our study regarding the lack of long-chain triglycerides (Intralipid) and Ropivacaine concentration measurement in the plasma of our animal subjects. However, the half-life of long-chain triglycerides in healthy rats was previously reported by Ge et al. (ref. 23) to be more than 30 minutes (0.63 h, more precisely). We added the following comment to our Discussions Section: “the time interval between LE and LA administration in our setup (2 minutes) was much shorter than the half-life of long-chain triglycerides, the main constituent of Intralipid administered in rats, which was reported to be more than 30 minutes (23). We can thus safely assume that most of the LE was still present in the circulation at the time Ropivacaine was injected. ”

“Minor Comments:

  1. The difference in HR between the two groups was trending significance. The authors should comment on this and perhaps improve the study design by increasing the number of rats studied to diminish this difference between groups.
  2. Similarly, the decrease in HR due to LE in Group B trended statistical significance. An increase in number of rats studied would help clarify this result.”

We agree that the p-values obtained by applying the t-tests for the comparisons mentioned by the reviewer (baseline heart rate comparison between groups A and B, p = 0.073; decrease in heart rate due to LE in group B, p = 0.085) are close to 0.05 and may be seen as “trending significance”. However, both values are greater than the significance threshold value set at the beginning of the study, before the measurements were performed. These values therefore indicate the acceptance of the null hypothesis in both cases, thus the interpretations given in the manuscript: - baseline values for heart rate in groups A and B were similar (p = 0.073, i.e. the difference between groups is not statistically significant) - In rats from group B, intravenous LE did not change the heart rate (p = 0.085, paired t-test; i.e. the change in heart rate after LE, compared to baseline in group B is not significantly different from 0). We believe that changing the study design (including more subjects) post hoc, hoping to obtain greater p- values (further away from the significance threshold 0.05) would be a methodological fault. The analysis was performed “in intention to treat”, as required by good statistical practices and the results were presented as such.

Reviewer 2 Report

In this study entitled “Cardio Protective effects of Lipid Emulsion against Ropivacaine-induced Local Anesthetic Systemic Toxicity – an experimental study” the authors investigated in an animal model the role of prophylactic injection of lipid emulsion to prevent and contrast the systemic toxicity induced by local anesthetics.

The topic is of interest, with interesting implications for surgeons, interventional cardiologists as well as for clinicians, and with unmet needs to be explored. The article is quite well written (even if several improvements are required), but the experimental method and results are to be more clearly presented. Moreover, the discussion has to be tempered to be concordant with the results. Here are my comments:

  1. In the abstract abbreviations should be avoided. If this is not possible, the authors must fully report the expression the first time they mention it and specify the abbreviation in brackets. What dose LA and LE stand for in abstract? This is valid also for the rest of the manuscript. Please, edit.
  2. The authors in introduction report that “Local anesthetics are chemicals that belong to one of the two categories: amides or esters.” As they choose to mention it, please provide a short description of the difference between these two categories, otherwise the information is just confounding and should be removed.
  3. “Although the occurrence of LAST is not very frequent, with a number of 459 cases being reported in a 5-year period…”. What do the authors mean? 459 of how many patients? Which population did they consider? The 5-year period is referred to what? Please better specify.
  4. “Among LE side effects, the ones worth mentioning are acute kidney injury, metabolic acidosis, acute lung micro embolism, or cardiac arrest.” The authors should provide the respective incidences for such complications. This is important so that the reader can have an idea of the risk/benefit ratio.
  5. The endpoints of the study should be clearly stated in section “Methods”. Moreover, they must clearly specify how they intended to assess the cardioprotective effect of LE.
  6. “To explore the possible protective role of lipid emulsion against the cardiac toxicity due to systemic administration of LA, 19 rats were randomly assigned into two groups: in the first group only intravenous LA Ropivacaine was administrated (group A); in the second group intravenous LE was used before Ropivacaine (group B)”. This should be moved to section “methods” and in the section “results” the authors should only mention how many subjects per group were included.
  7. “Microscopy findings” and “Figure 3”. In this part, is not fully clear what the authors mean by “control group”, never mentioned before. Is this another group of subjects (and in this case this has to be specified in section methods)? Moreover, how could they obtain histological sections D, E and F (lipid administration only)? This was another group? Or some of the subjects in group B were sacrificed before receiving Ropivacaine (and in this case this has to be specified in section methods)? Or what else? Please specify in detail.
  8. The authors suggest to administer lipid emulsion prior Ropivacaine. In a clinical setting one would expect the lipid emulsion to be administered after an accidental administration of intravascular LA, as a kind of antidote and not as a prophylactic agent (otherwise one should always administer it before any procedure). The results of their study, despite show a certain trend to cardioprotective effect of LE in preventing LAST, are far from being sufficient to suggest a routinary prophylactic use of LE in all procedures requiring LA. This must be clearly stated, discussed and recognized by authors in section discussion, that has to be modified accordingly.
  9. The overall English style of the text could be improved. For example, there are several cases of repetitions of the same word in a single sentence.

Author Response

Thank you very much for the insightful observations.
Here are our answers to your requests on the manuscript. They can also be tracked on the manuscript.
1. In the abstract abbreviations should be avoided. If this is not possible, the authors must fully report the expression the first time they mention it and specify the abbreviation in brackets. What dose LA and LE stand for in the abstract? This is valid also for the rest of the manuscript. Please, edit
We introduced the explanations for the abbreviations. The dosage of Local Anesthetic (LA) and the Lipid Emulsion (LE) are, according to reference no 15 which is a publication on the method we conducted this study. For the LA we used 0.1 ml and for the LE 0.3- 0.4 ml was used based on the  calculation method we described in reference 15 and included also in the Method section. These data were included in the abstract and the methods section as well.
2. The authors in the introduction report that “Local anesthetics are chemicals that belong to one of the two categories: amides or esters.” As they choose to mention it, please provide a short description of the difference between these two categories, otherwise, the information is just confounding and should be removed.
We removed the mention of the amides or esters chemical classes. Instead, we associated a summary of the chemical structure of LA to the possible mechanism of action of LE in LAST (please see the fifth paragraph of the Introduction section).
3. “Although the occurrence of LAST is not very frequent, with a number of 459 cases being reported in a 5-year period…”. What do the authors mean? 459 of how many patients? Which population did they consider? Please better specify.
This example refers to the results from The National Poison Data System (NPDS) -USA captures data on all calls to poison centers serving the entire United States and its territories. 
The reference presents a review that presents the data extracted from NPDS registries over a 5-year period in adult people.
4. “Among LE side effects, the ones worth mentioning are acute kidney injury, metabolic acidosis, acute lung micro embolism, or cardiac arrest.” The authors should provide the respective incidences of such complications. This is important so that the reader can have an idea of the risk/benefit ratio.
The incidence of the LE complications islisted in the manufacturer’s marketing release as less frequent (less than 1%). We inserted this information in the manuscript with the assigned reference.
5. The endpoints of the study should be clearly stated in section “Methods”. Moreover, they must clearly specify how they intended to assess the cardioprotective effect of LE. 
The cardioprotective effect of LE 20% was assessed by rhythm analysis of the cardiac tracings of the subjects. A protective effect was considered when the normal cardiac rhythm was maintained longer or the EKG components (length of the QRS complexes or the heart rate) were maintained after Ropivacaine was administered.
6. “To explore the possible protective role of lipid emulsion against the cardiac toxicity due to systemic administration of LA, 19 rats were randomly assigned into two groups: in the first group only intravenous LA Ropivacaine was administrated (group A); in the second group intravenous LE was used before Ropivacaine (group B)”. This should be moved to section “methods” and in the section “results” the authors should only mention how many subjects per group were included.
The suggested changes were made in the manuscript.
7. “Microscopy findings” and “Figure 3”. In this part, is not fully clear what the authors mean by “control group”, never mentioned before. Is this another group of subjects (and in this case this has to be specified in section methods)? Moreover, how could they obtain histological sections D, E and F (lipid administration only)? This was another group? Or some of the subjects in group B were sacrificed before receiving Ropivacaine (and in this case, this has to be specified in section methods)? Or what else? Please specify in detail.
Specifications were made in the methods section, and we also provide them here.
The subjects were divided into 4 groups: 
Group A – (n=10) – LA Ropivacaine 2mg/ml 0.1 ml was administered every 2 minutes and cardiac activity was monitored after 4 administrations tissue samples were harvested from the kidney, heart, and liver were harvested.
Group B – (n= 9) – LE 20% 0.3-0.4 ml was administered 2 minutes prior 0.1ml Ropivacaine 2mg/ml. Cardiac activity was monitored and after 4 administrations of Ropivacaine the subject was sacrificed, and tissue samples were harvested from the kidney, heart, and liver were harvested
Group C – (n=8) which received just Lipid Emulsion 20% under general anesthesia, cardiac activity was monitored and tissue samples from kidney, heart, and liver were harvested
Group D- control group (n= 8) which received none of the studied substances and were anesthetized, the cardiac activity was monitored and tissue samples from kidney, heart, and liver were harvested.
8. The authors suggest to administer lipid emulsion prior Ropivacaine. In a clinical setting 
one would expect the lipid emulsion to be administered after an accidental administration of intravascular LA, as a kind of antidote and not as a prophylactic agent (otherwise one should always administer it before any procedure). The results of their study, despite show a certain trend to cardioprotective effect of LE in preventing LAST, are far from being 
sufficient to suggest a routinary prophylactic use of LE in all procedures requiring LA. This must be clearly stated, discussed and recognized by authors in section discussion, that has to be modified accordingly.
The main objective of this study is to observe if LE can be administered in a prophylactic fashion, to avoid the prospective of LAST. At this moment indeed LE is used in LAST as a treatment, but we would like to test if it can also be used as prophylaxis to reduce even more the incidence of this deadly medical emergency, LAST.
Our study results accomplish to reach its objective, that LE 20% has the possibility of being used as well as prophylaxis in Ropivacaine induced LAST, not only as a treatment for an already diagnosed toxicity. 
We added a statement in the Discussion section in this respect.
“In present clinical practice, LE is only used in LAST as a treatment, after the onset of symptoms. The novelty of our study is the introduction of LE as a prophylactic agent in Ropivacaine-induced LAST, instead of a treatment for an already diagnosed toxicity. Although not sufficient for supporting the routine prophylactic use of LE in all procedures requiring LA, our results are encouraging, showing a cardioprotective effect and negligible toxicity of LE in our experimental model. Further safety and efficacy studies are warranted to explore the benefit of this prophylactic approach.”
9. The overall English style of the text could be improved. For example, there are several cases of repetitions of the same word in a single sentence
We sought English proofing by a native English speaker, and we made changes throughout the manuscript

Round 2

Reviewer 2 Report

The authors provided a revised version of the manuscript with all the required changes. This resulted in a significant improvement of the overall quality of their paper. I have no further observations.